# Equatorial Atlantic interannual variability and its relation to dynamic and thermodynamic processes

Julien Jouanno[1], Olga Hernandez[2,1], and Emilia Sanchez-Gomez[3]

[1]LEGOS, Université de Toulouse, CNES, CNRS, IRD, UPS, Toulouse, France.

[2]Mercator-Océan, Ramonville Saint Agne, France

[3]CECI/CERFACS, Toulouse, France

*Correspondence to*: Julien Jouanno (julien.jouanno@ird.fr)

**Abstract.** The contributions of the dynamic and thermodynamic forcing to the interannual variability of the Equatorial Atlantic sea surface temperature are investigated using a set of interannual regional simulations of the Tropical Atlantic Ocean. The ocean model is forced with an interactive atmospheric boundary layer, avoiding damping toward prescribed air-temperature as is usually the case in forced ocean models. The

model successfully reproduces a large fraction ($R^2$=0.55) of the observed interannual variability in the Equatorial Atlantic. In agreement with leading theories, our results confirm that the interannual variations of the dynamical forcing largely contributes to this variability. We show that mean and seasonal upper ocean temperature biases, commonly found in fully coupled models, strongly favor an unrealistic thermodynamic control of the Equatorial Atlantic interannual variability.


## 1. Introduction

The main mode of interannual variability in the Tropical Atlantic is generally referred as Atlantic Equatorial Mode or Atlantic Niño [Zebiak 1993, Richter et al. 2013]. It consists in anomalies of sea surface temperature (SST) along the equator with largest variability during boreal summer (Jun-Jul-Aug; JJA) and

a spatial extent that covers the cold tongue area [e.g., see Lübbecke and McPhaden 2013].

Many observational or modeling studies suggested that wind forced ocean wave dynamics play a crucial role in controlling the equatorial Atlantic interannual variability. Early work by Hirst and Hastenrath [1983] or Servain et al. [1982] show evidences of remote forcing of eastern SST anomalies by Atlantic zonal winds in the western part of the basin, the link between the two regions being provided by

the propagation of equatorial Kelvin waves and their influence on the equatorial thermocline depth. Using

observations and intermediate complexity coupled model, Zebiak [1993] suggested that the delayed oscillator mechanism is the main mechanism underlying the oscillating interannual variability in the Atlantic. Ocean dynamics are implicit to the delayed oscillator mechanism: Rossby and Kelvin waves provide the phase-transition mechanism for the oscillator cycle. The analysis of reanalysis products by

Lübbecke and McPhaden [2017] confirms that the Bjerknes feedback is operative in the Tropical Atlantic [Keenlyside and Latif, 2006]. The mentioned studies show a thermocline depth-SST relationship in the eastern part of the equatorial Atlantic that is as strong or even stronger than for the Pacific. The analysis by Foltz and Mc Phaden [2010] and Burmeister et al. [2016] also revealed the key role of the reflection of planetary Rossby wave into an equatorial Kelvin in preconditioning, through thermocline rising, an

anomalously strong surface cooling in the Atlantic cold tongue area during summer 2009. Planton et al. [2017] also highlight the importance of eastward propagating equatorial Kelvin wave, advection and mixing in controlling the interannual variability of the central equatorial temperatures.

Hence, our present understanding of the equatorial Atlantic interannual variability involves a large contribution of ocean dynamics. However, results obtained in recent studies questioned this paradigm. On

the basis of the analysis of two contrasted warm (2002) and cold tongue events (2005), Hormann and Brandt [2009] found a weak impact of the equatorial Kelvin wave on the SST of the equatorial Atlantic. Richter et al. [2012] show that some equatorial Atlantic warm events are not explained by equatorial dynamics, but are due to horizontal advection of off equatorial warm temperature anomalies. More recently, the analysis by Nnamchi et al. [2015] from a set of CMIP5 simulations including full coupled

global circulation model (GCM) and slab GCM suggests that the Atlantic Niño variability, as resolved by state-of-the-art coupled models, mainly depends on the thermodynamic component ($R^2$=0.92).

Despite a growing understanding of the processes involved in the control of equatorial Atlantic interannual variability, these recent studies challenge its functioning and ask for a better quantification of the relative contributions of the thermodynamic and dynamic forcing, and how both contributions are

represented in models comparing to observations. This is the main objective of our study. We will examine how much the ratio between the two contributions depends on the upper ocean seasonal bias generally found in coupled models. The paper is organized as follows. Section 2 presents the simulation strategy. Section 3 is dedicated to quantify the dynamic and thermodynamic contributions using mixed-layer heat budget in long term simulations (1979-2015), together with comparison between simulations forced with

climatological or interannually varying wind stress. Section 4 discusses how seasonal biases impact the equatorial response to interannual anomalies of the atmospheric forcing. Conclusions are given in Section 5.

## 2. Simulations and data

### 2.1 The regional configuration

The numerical code is the oceanic component of the Nucleus for European Modeling of the Ocean program [NEMO3.6, Madec, 2016]. It solves the three dimensional primitive equations in curvilinear coordinates discretized on a C-grid and fixed vertical levels (z-coordinate). The model configuration consists of a grid with 1/4° horizontal resolution ($\Delta x$, $\Delta y$ ~ 25km) encompassing the Equatorial Atlantic (from 60°W to 15°E and from 20°S to 20°N; see model domain in Figure 1). There are 75 levels on the vertical with 12 levels in the upper 20 meters and 24 levels in the upper 100 meters. Temperature and salinity are advected using a Total Variance Dissipation scheme (TVD) with nearly horizontal diffusion parameterized as a Laplacian isopycnal diffusion, with a coefficient of 300 $m^2$ $s^{-1}$. Horizontal diffusion of momentum is implicit since a third order advection scheme UP3 is employed. The vertical diffusion coefficients are given by a Generic Length Scale (GLS) scheme with a *k-ε* turbulent closure. Bottom friction is quadratic with a bottom drag coefficient of $10^{-3}$ and partial slip boundary conditions are applied at the lateral boundaries. The free-surface is solved using a time-splitting technique with the barotropic part of the dynamical equations integrated explicitly.

Horizontal velocity, temperature, salinity and sea level are specified at the lateral boundaries of model domain using climatological conditions computed from 1992-2012 daily outputs of the MERCATOR global reanalysis GLORYS2V3 [Ferry et al. 2012]. Very similar configurations of this regional setup, using earlier version of the NEMO model, have been used to investigate mechanisms of variability of the SST [Jouanno et al. 2011, Jouanno et al. 2013] or sea surface salinity [Da Allada et al, 2017] in the Tropical Atlantic.

### 2.2 Surface forcing strategy

At the surface, the air-sea fluxes of momentum, heat and freshwater are computed using bulk formulae [Large and Yeager, 2009]. The specification of atmospheric conditions (air temperature, humidity, and wind speed) when forcing an ocean model with bulk formulae acts to restore the SST toward prescribed air-temperature. The method constrains the model solution toward further realism, but as a main drawback, a realistic representation of the SST interannual variability cannot guarantee that the correct processes are at play in the model. This damping also prevents the use of sensitivity experiments to evaluate the impact of the interannual variability of atmospheric variables other than the air-temperature.

To partly overcome this issue, the evolution of the atmospheric boundary layer temperature and humidity are computed with the simplified atmospheric boundary layer model CheapAML [Deremble et al. 2013], letting the wind field to be prescribed. The model consists of two prognostic equations for

atmospheric temperature and humidity. The fraction of humidity entrained at the top of the atmospheric boundary layer is taken a 0.25 following Seager et al. [1995]. The boundary layer height is prescribed using the planetary boundary layer height climatology derived from ERA-Interim re-analysis from Von Engeln and Teixeira [2013].

## 2.3 Simulations

We carried out four simulations referred henceforth as REF, REF-$\tau_{clim}$, BIASED, and BIASED-$\tau_{clim}$. The model reference simulation (REF) is forced with DFS5.2 [Drakkar Forcing Set; Dussin et al. 2016] which is based on corrected ERA-interim reanalysis fields, and consists of 3-hour fields of wind and daily fields of long, short wave radiation and precipitation. The shortwave radiation forcing is modulated on-line by a theoretical diurnal cycle.

Experiment REF-$\tau_{clim}$ is forced with monthly climatological wind stress. The modified wind stress is used as boundary condition for both the momentum equations and the vertical turbulence closure scheme, but the surface fluxes of heat and freshwater remain forced by the interannual data. This strategy allows to specifically remove the dynamical contribution of the interannual winds. However, thermodynamic contributions of wind variability (i.e. latent and sensible heat) are allowed to vary interannually.

A second set of simulations (BIASED and BIASED-$\tau_{clim}$) has been performed, in which the seasonal cycle of the prescribed atmospheric variables (wind, long wave radiation, short wave radiation and precipitation) is replaced by the seasonal cycle simulated by a coupled model. The biased seasonal cycle we used in this study is issued from an ensemble of 10 members performed with the CNRM-CM5 model for the period 1979-2012 [Voldoire et al. 2011]. The seasonal cycles have been isolated using harmonic analysis and the CNRM-CM5 data were interpolated on the DFS5.2 grid. This ensemble belongs to the 20[th] century historical experiment available in the CMIP5 (Coupled Model Intercomparison Phase 5) dataset. CNRM-CM5 model exhibit a marked equatorial Atlantic warm SST bias typical of the CMIP5 ensemble mean warm bias [Richter et al. 2008, Voldoire et al. 2014]. Similarly to REF and REF-$\tau_{clim}$, a set of two ocean stand-alone simulations: referred as BIASED (forced by the interannual forcing biased to CNRM-CM5 climatology), and BIASED-$\tau_{clim}$ (forced with biased climatological wind stress from CNRM-CM5) have been performed.

All the simulations are run from 1958 to 2015 and daily means from 1979 to 2015 are analysed in this study.

## 2.4 Model mixed-layer heat balance

The mixed layer heat content equation can be written as (cf Menkes et al. [2006] or Jouanno et al. [2011]):

$$\underbrace{< \partial_t T >}_{\text{TOT}} = \underbrace{-< u\,\partial_x T > -< u\,\partial_y T > +< Dl(T) >}_{\text{HOR}}$$

$$\underbrace{-< w\,\partial_z T > -\frac{1}{h}\frac{\partial h}{\partial t}(< T > - T_{z=-h}) + \frac{1}{h}(Kz\,\partial_z T)_{z=-h}}_{\text{VER}} + \underbrace{\frac{Q*+Qs\,(1-f_{z=-h})}{\rho_0 C_p h}}_{\text{FOR}},$$

with

$$<\cdot> = \frac{1}{h}\int_{-h}^{0} \cdot\,\partial z,$$

with T the model potential temperature, (u, v, w) are the velocity components, Dl(T) the lateral diffusion operator, Kz the vertical diffusion coefficient for tracers, and h the mixed layer depth. Here, Q* and Qs are

respectively the non penetrative (latent, sensible and longwave heat fluxes) and penetrative components of the air–sea heat flux (shortwave radiation), and fz=-h is the fraction of the shortwave radiation that reaches the mixed layer depth (MLD). The MLD is defined as the depth where the density increase compared to density at 10 m equals 0.03 kg m$^{-3}$. TOT represents the total mixed-layer temperature tendency, HOR the tendency associated with horizontal processes including advection and lateral diffusion, VER the tendency

associated with vertical processes including the vertical advection, the turbulent flux at the base of the mixed layer, and the mixed layer temperature variations due to the displacements of the mixed layer base, and finally FOR is the air–sea heat flux storage in the mixed layer. This equation will be used to diagnose the origin of the heat content variations in the mixed layer in our simulations.

## 2.5 Observations

The four simulations will be compared to several observational products. Observations for SST and air-sea fluxes are from TropFlux [Praveen Kumar et al. 2012]. Data are available for the period 1979-2015 and are based on bias and amplitude corrections from ERA-interim and ISCCP shortwave data. Correction were performed on the basis of comparison with the Global Tropical Moored Buoy Array, so there are specific to the tropical region. Monthly fields of ISAS-13 temperature [Gaillard et al., 2016], available for the period

2004–2012 at 1/4° spatial resolution, are also used. In the Tropical Atlantic, they consist of an optimal interpolation of observations from Argo profiling floats and PIRATA moorings.

## 3.  Dynamic vs thermodynamic control of the interannual variability

A key feature of the Tropical Atlantic variability is the so called Atlantic cold tongue (Figure 1a), whose extension and strength peaks in June-July-August (JJA), as a consequence of seasonal surface cooling driven by subsurface processes that develops along and south of the equator [e.g., Wade et al. 2011, Jouanno et al. 2011]. In JJA, the mean SST in REF compares well with SST from TropFlux (Figure 1b).

There is a warm bias (~1.5°C) located in the southern hemisphere along the African coast, but it is weak compared to the warm bias found in state-of-the-art coupled models which can annually exceed 5°C [e.g., Voldoire et al. 2014, Richter et al. 2008, 2012]. At the equator, the seasonal formation of the cold tongue is well reproduced in the REF simulation, with an equatorial bias lower than 1°C (Figure 1b). At subsurface, the observed east-west tilt of the thermocline (Figure 1d) is well reproduced in REF (Figure 1e) however

the ocean model thermocline is not as sharp as in the observations. This is a long lasting problem of equatorial modelling. Possible implications of this bias for the representation of the interannual variability of the surface temperature are out of the scope of this study but would deserve further attention.

The year to year evolution of JJA SST averaged over Atl3 (between 20°W-0° and 3°S-3°N as defined in Zebiak 1993) is shown in Figure 2a. REF simulation reproduces well the amplitude of the SST

present in the observations (regression slope between REF and observations; a=0.86) and also a substantial fraction of the observed interannual variability ($R^2$=0.55), with most of the model Atlantic Niño and Niña events in phase with observations. The removal of the interannual variability of the wind stress in REF$\tau_{clim}$ clearly reduces the amplitude (a=0.26 and $R^2$=0.27 between REF and REF$\tau_{clim}$) of the interannual variability (Figure 2a) and weakens the correlation with the observations ($R^2$=0.25). This suggests that the

interannual dynamical forcing actively participates to the control of the Atlantic Niño and Niña events. The interannual variance of Atl3 shows a marked seasonal cycle with maximum of variance in May-June-July in the observations (Figure 2b). This seasonal cycle is slightly shifted in REF, with interannual variability in March-April stronger than the observed interannual variability. More interestingly, we note that the interannual standard deviation is drastically reduced when removing the interannual variability of

the wind stress (REF$\tau_{clim}$ Figure 2b). This suggests that the dynamical component of the interannual forcing is not only active in summer but also all along the year.

In order to get further insight on the nature of the dynamical processes at play during the warm and cold events, we performed a composite analysis of 8 Niño and 7 Niña years selected over the period 1979-2015. Following the methodology proposed in Lübbecke and McPhaden [2017], the Niño and Niña years

are selected when detrended interannual SST anomalies from TropFlux averaged over Atl3 exceed the standard deviation of the time series for at least 2 months between May and September. Lübbecke and McPhaden [2017] has shown large symmetry of the Niño and Niña events in the Atlantic in terms of development and processes, so we will focus on the anomalies between both type of events.

In both model and observations, the seasonal evolution of the SST in Atl3 during Niño and Niña years, indicate that in average the temperature anomalies form early in the season (in March-April) and start to vanish from August-September (Figure 3a). This is consistent with findings by Lübbecke and McPhaden [2017]. There is a large (~30 W m$^{-2}$) difference between model and TropFlux estimates of the

mean net air-sea heat flux at the sea surface (Figure 3b). However, this difference is within the range of the differences found between state-of-the-art air-sea heat flux products in equatorial cold tongue regions (e.g. see Fig. 16 of Praveen-Kumar et al. 2012 for Nino3). Most importantly, both model and observation show that the net heat flux acts toward a reduction of the temperature anomalies from May to August. This further indicates that the thermodynamic forcing is not the leading mechanism to explain the interannual

variability of Atl3 in JJA.

The analysis of the mixed-layer heat balance indicates that the vertical subsurface processes control the occurrence of Niño or Niña events, in addition to cool the mixed-layer of Atl3 all year long (Figure 3c). In contrast and as noted earlier in Planton et al. [2017], the warming by air-sea fluxes (FOR) and horizontal advection (HOR) is increased during cold events and reduced during warm events, so these processes act to

reduce the temperature anomalies. Anomalous subsurface cooling is achieved by anomalous vertical diffusion of heat (Figure 3f), in response to anomalous thermocline depth (Figure 3e), as also noticed by Planton et al. [2017]. The anomalies of thermocline depth form early in the season (January-February-March) but the largest anomalies of vertical diffusion occur in May-June-July. This apparent contradiction is easily reconciled when considering that May-June-July is a period more prone for thermocline depth

anomalies to bring their imprint on the surface temperature. Indeed, during this period: i) the thermocline is getting closer to the surface (Figure 3e) and above all, and ii) the westward surface current is intensified (Figure 3d), providing an efficient source of shear driven turbulence between the mixed-layer and the thermocline below [e.g. see Jouanno et al. 2011]. Interannual anomalies of the surface currents could also participate to anomalies of vertical diffusion by increasing the levels of turbulence with the Equatorial

Undercurrent below, but the lack of agreement between anomalies of zonal surface velocity (Figure 3d) and anomalies of vertical diffusion (Figure 3f) suggest they have not a first order influence.

Spatial maps of correlation between time series of season average surface temperatures from REF and REF-$\tau_{clim}$ are shown in Figure 4. Values of $R^2$ close to 1 suggest that thermodynamic processes play a dominant role on the interannual variability of the SST, while values close to 0 suggest that the variability

is controlled by the dynamical component. The correlation maps indicate that the interannual variability of the SST in the equatorial and coastal upwelling areas is mainly controlled by dynamics while in the subtropical gyres thermodynamic play a dominant role. At the equator, the influence of the dynamics is

larger in JJA and SON, most probably due to shallow thermocline and intensified Tropical Wave Instability respectively.

## 4.  Impact of seasonal biases on Atlantic Niños and Niñas representation

Our results are at odds with the results by Nnamchi et al. [2015] suggesting a thermodynamic control of the equatorial interannual SST variability. Most of the CMIP5 models simulate a warm bias at the Equator [Richter et al. 2012], and our hypothesis is that such bias deeply modifies the response to interannual winds in such a way it favors a thermodynamic response. To test this hypothesis we analyzed our set of simulations forced with biased atmospheric variables issued from the CNRM-CM5 coupled model
(BIASED and BIASED-τclim).

The simulation BIASED reproduces a warm bias in the cold tongue area that reaches 7°C near the African coast in JJA (Figure 1c) with a spatial structure typical of the bias found in coupled models in the region [e.g. Richter et al, 2014]. The annual mean bias (not shown) reaches 5°C and resembles the bias of CNRM-CM5 coupled model [Voldoire et al. 2014]. In BIASED, there is no more east-west tilt of the
thermocline (Figure 1d), and the thermocline is even more diffuse than in REF. BIASED is forced with the same interannual anomalies of winds, downward radiative fluxes and precipitation as in REF, but the interannual responses of the surface temperature of the two simulations are very different. First, the SST interannual variability is no more correlated with TropFlux (Figure 2a; $R^2$=0.02). Second, the maximum of variance is shifted toward boreal spring (Figure 2b). This highlights how much the interplay between
interannual anomalies and the seasonal variability is critical in the functioning of the interannual Atlantic equatorial variability.

Unlike the results obtained from the reference simulations ($R^2$ between REF and REF-$\tau_{clim}$ of 0.27), removing the interannual variability of the wind stress in BIASED has a much weaker impact on the interannual variability of Atl3 in JJA as shown by the comparison between BIASED and BIASED-$\tau_{clim}$ in
Figures 2a,b ($R^2$=0.77). This high correlation between BIASED and BIASED-$\tau_{clim}$ suggests that thermodynamic processes mainly drive the equatorial interannual variability in BIASED. This is confirmed by the mixed-layer heat balance of Niño and Niña events (Figure 5) that illustrates how the interannual variability is driven almost entirely by the air-sea heat fluxes, with the subsurface vertical processes now damping the Niño and Niña anomalies. The seasonal correlation maps indicate that the dynamics control of
the interannual SSTs in BIASED in the equatorial and coastal upwelling areas is reduced for all the seasons and is almost absent in DJF and MAM (Figure 6).

## 5.  Discussion and conclusion

The objective of this study was to clarify the role of the dynamical processes in controlling the interannual variability of the Tropical Atlantic SSTs, and how they are represented in ocean stand-alone and fully coupled models. For the stand-alone ocean model, we overcome the difficulties inherent to the use of a forced ocean model when analyzing processes of interannual variability, by coupling the ocean model with an atmospheric boundary layer model that provides interactive air temperature and humidity. In addition to a better representation of the air-sea exchanges, such strategy allowed to properly assess the sensitivity of the interannual variability of the equatorial Atlantic surface temperature to the interannual variability of the equatorial wind stress.

The recent study by Nnamchi et al. [2015] downplayed the role of the dynamics in controlling the interannual variability of the Atlantic Niños and Niñas. Instead, our results suggest that ocean dynamics indeed control a large fraction of the equatorial SST interannual variability, in full agreement with recent results by Planton et al. [2017]. This is also in line with early and more recent studies suggesting that coupled equatorial dynamics play an important (but not exclusive) role on the equatorial Atlantic interannual variability [Zebiack 1993, Lübbecke and McPhaden 2017]. Moreover, we showed that a biased atmospheric forcing issued from a coupled model simulations deeply modifies the oceanic heat budget and its response to interannual anomalies of air-sea fluxes of heat and momentum. This strongly suggests and confirms that even if the Atlantic Equatorial Mode is represented in state-of-the-art coupled models, the dynamical oceanic processes are underestimated, while the thermodynamic processes are the main driver of the variability. This fact is likely due to strong biases in the atmospheric component, that induce an incorrect ocean circulation and its associated variability [Richter et al, 2008, 2012].

Our results further illustrate how the interplay between interannual anomalies of the surface forcing and the seasonal variability is key to interpret equatorial Atlantic variability. The thermocline anomalies during Niño or Nina years form early in the season (Jan-Feb-Mar) but the anomalies of the subsurface vertical heat flux at the base of the mixed-layer are at their largest in May-June-July, when seasonal turbulent mixed-layer cooling is at its maximum. This is in agreement with results by Burls et al. [2012] suggesting that the interannual variability in the equatorial Atlantic can be seen as a modulation of the seasonal cycle.

From a climate modelling perspective, and although a set of fully coupled simulations would be required to confirm our findings, our results are suggestive that a reduction of the mean and seasonal model biases in the Tropical Atlantic (in particular from the atmospheric component) would strongly benefit to the representation of the interannual variability. The variability of the Atlantic Cold Tongue exerts a significant influence on the climate of the surrounding regions and more specifically on the West African monsoon [Okumura and Xie 2004, Caniaux et al. 2011] or on rainfall variability in the

northeast of Brazil [Kushnir et al., 2006]. In terms of predictability at seasonal time scale of these phenomena, our results suggest that the ability of the climate models to maintain a realistic stratification and east-west tilt of the thermocline is key in correctly representing the response of the summer coupled system to spring wind anomalies. We also anticipate that a good representation of the dynamical processes in the Atlantic equatorial region will have an impact on the dynamics of the meridional overturning circulation (MOC) in coupled climate models. It seems that the MOC is indeed very sensitive to equatorial processes such as precipitation biases [Liu et al. 2017]: excessive precipitations near the equator tend to over stabilize the MOC, which is often an issue when trying to assess the stability of climate change scenarios.

**Acknowledgment**

*This study was supported by EU FP7/2007-2013 under Grant Agreement No. 603521, project PREFACE. Computing facilities were provided by GENCI project GEN7298. A special thank to Bruno Deremble for a careful reading of the manuscript and his assistance in porting and tuning the interactive atmospheric boundary layer CheapAML. Finally, we are grateful to the three anonymous reviewer for helpful comments on the manuscript.*

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

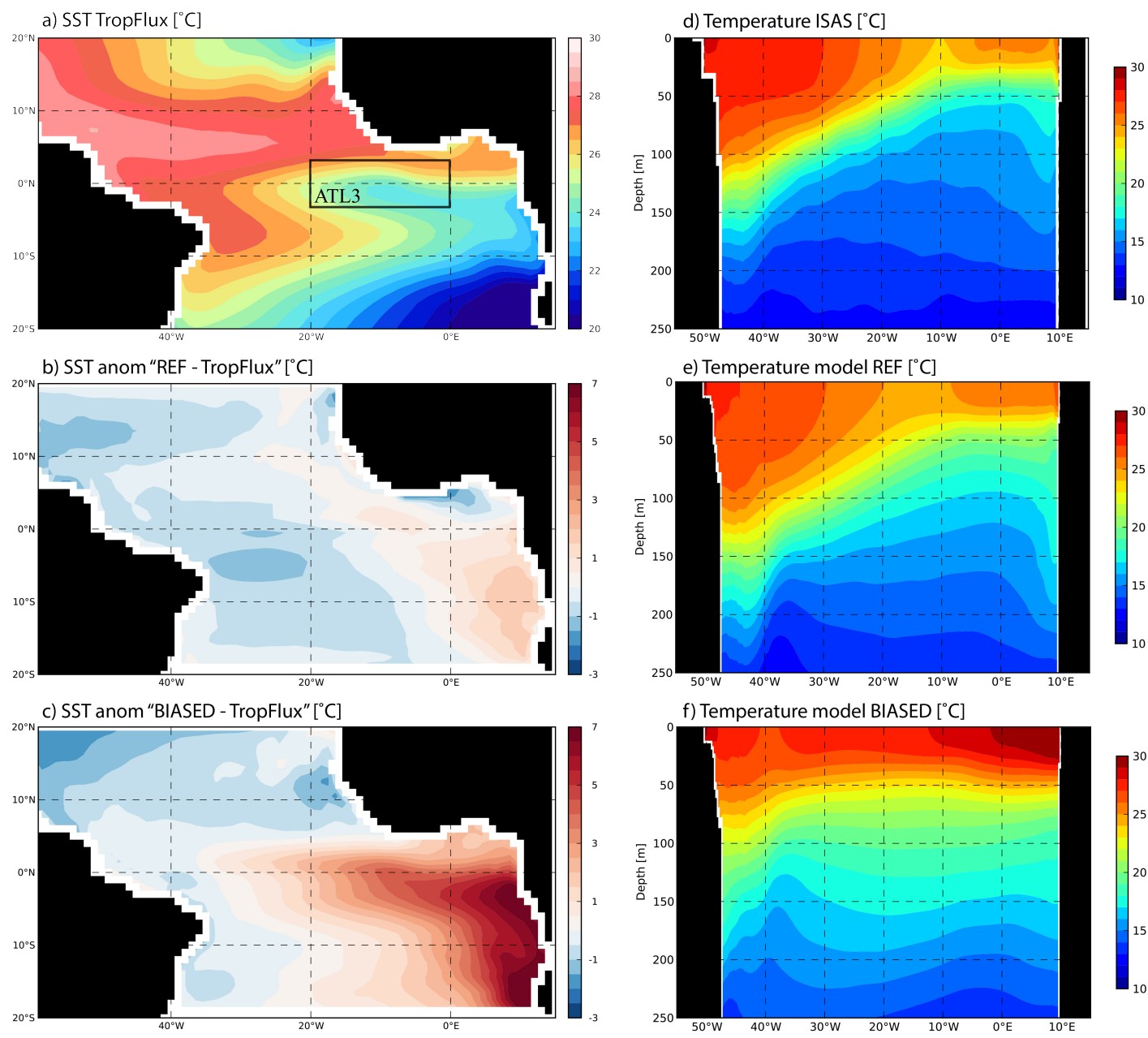

**Figure 1**. Climatological SST (°C) in June-July-August for the period 1979-2015 from TropFlux (a) and anomalies of SST between simulation REF and TropFlux (b) and between simulation BIASED and TropFlux (c). Zonal sections of June-July-August temperatures (°C) averaged between 2°S and 2°N from ISAS observations (d), REF (e) and BIASED (f).

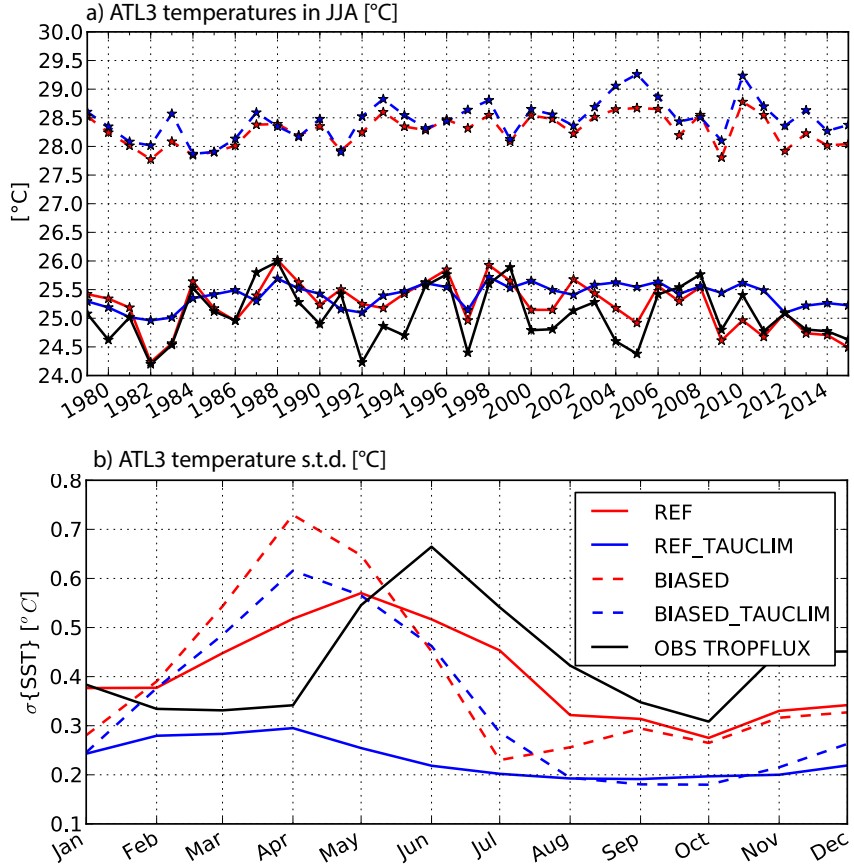

**Figure 2.** a) Time serie of Atl3 index (SST averaged in JJA between 20°W-0°N and 3°S-3°N) obtained from TropFlux and simulations. Monthly standard deviation of Atl3 SST using data from 1979 to 2015. Units are °C.

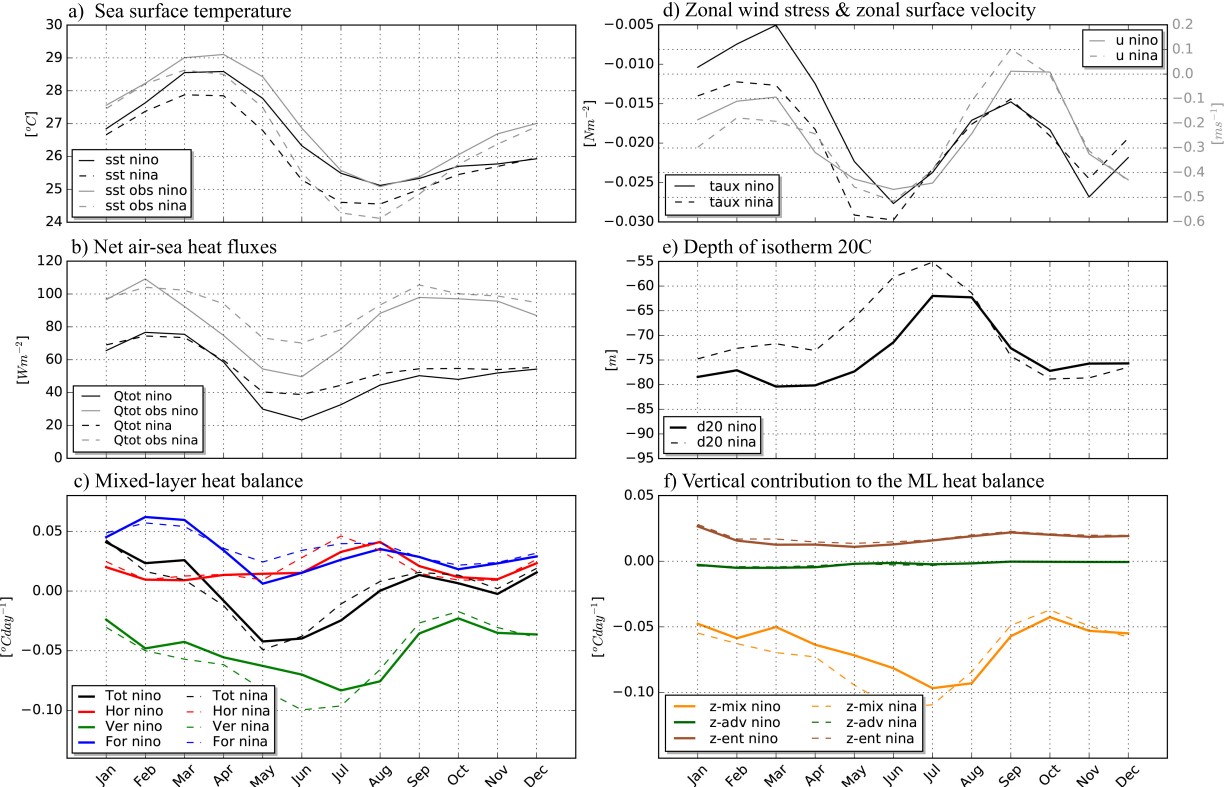

**Figure 3.** Composite seasonal evolution of variables from REF and observations averaged over Atl3 for Niño (continuous lines) and Niña years (dashed lines): a) surface temperatures from model and TropFlux, b) net air-sea heat flux from model and TropFlux, c) mixed-layer heat budget contributions as defined in Eq. 1, d) model zonal wind stress and zonal surface current, e) depth of the isotherm 20°C, f) mixing, advection and entrainment contributions to the mixed-layer contribution VER. Niño and Niña years were selected using SST from TropFlux using the methodology described in Lübbecke and McPhaden [2017].

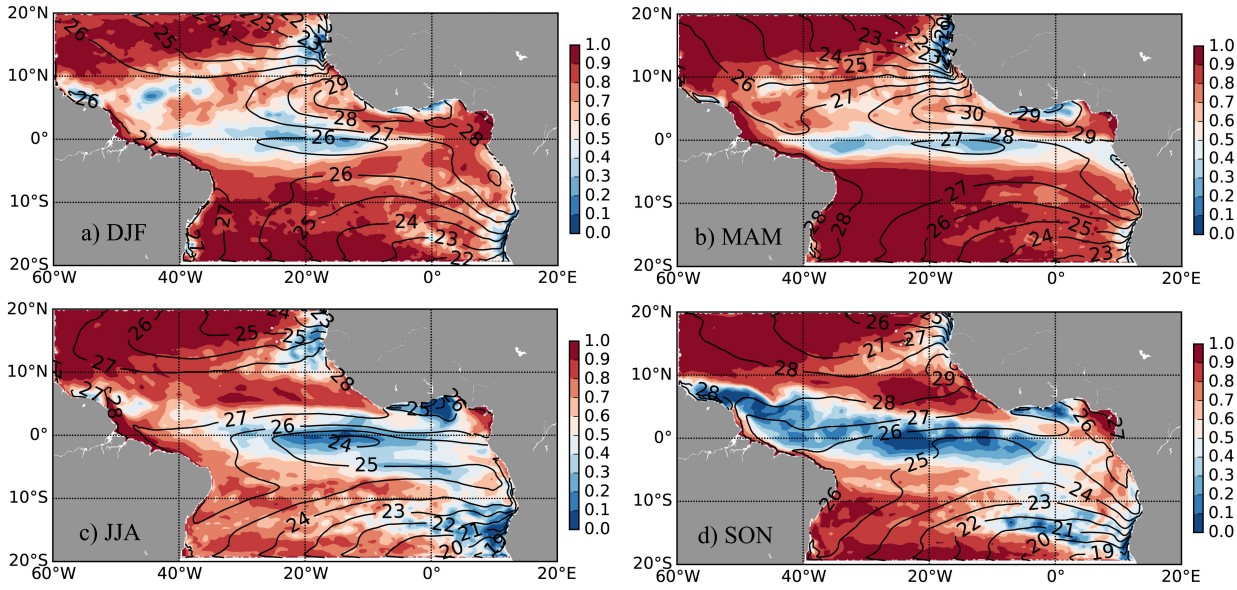

**Figure 4.** Coefficient of determination ($R^2$) between REF and REF-$\tau_{clim}$ seasonal SST time series. $R^2$ has been computed at each model grid point using data from 1979-2015 and with temperatures averaged over four seasons: Dec-Jan-Feb (a), Mar-Apr-May (b), Jun-Jul-Aug (c) and Sep-Oct-Nov (d). Values close to 1 indicate that seasonal SSTs in the two simulations are highly correlated, suggesting a thermodynamic control of the interannual variability, while values close to 0 indicate that seasonal SSTs in the two simulations are uncorrelated, suggesting a dynamic control of the interannual variability.

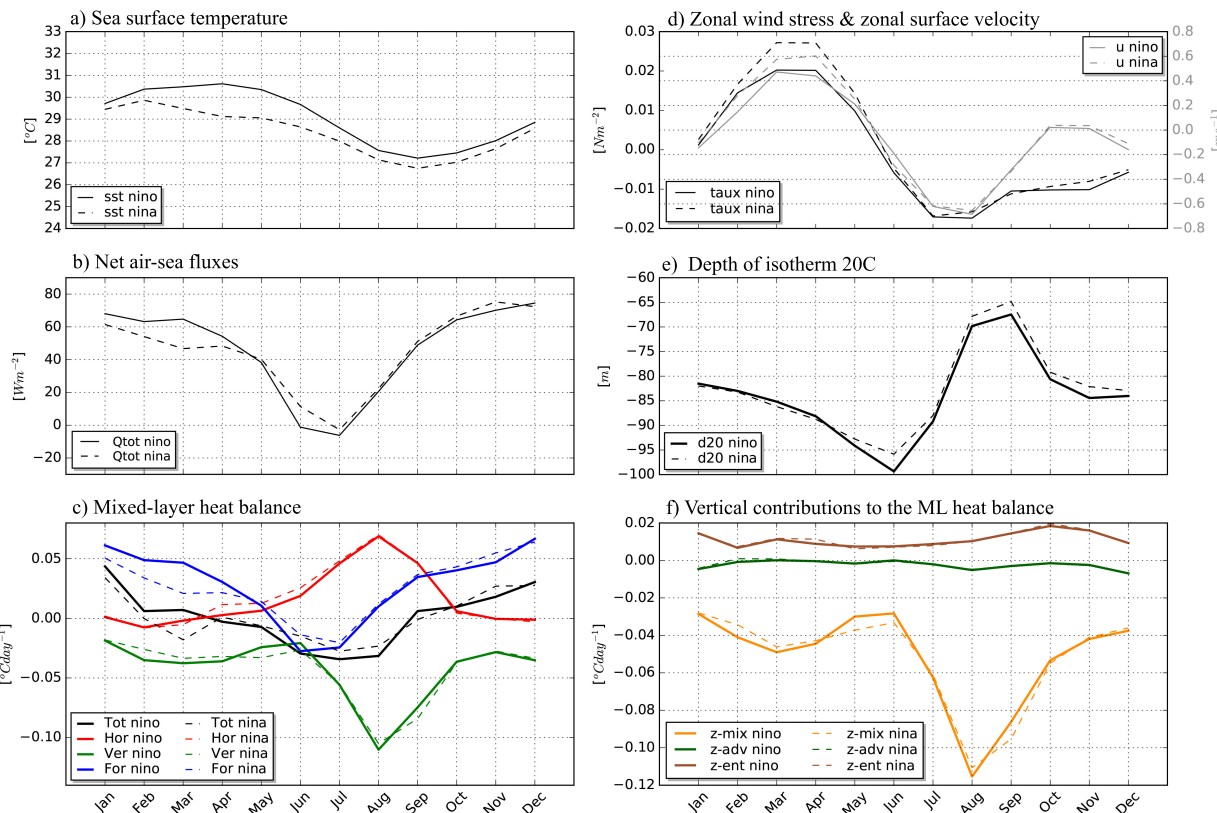

5  **Figure 5.** As Figure 3 but using BIASED simulation. Here, Niño and Niña years were selected following the same method as in Figure 3 but using Atl3 SSTs from BIASED so they do not necessarily coincide with observed Niño and Niña years.

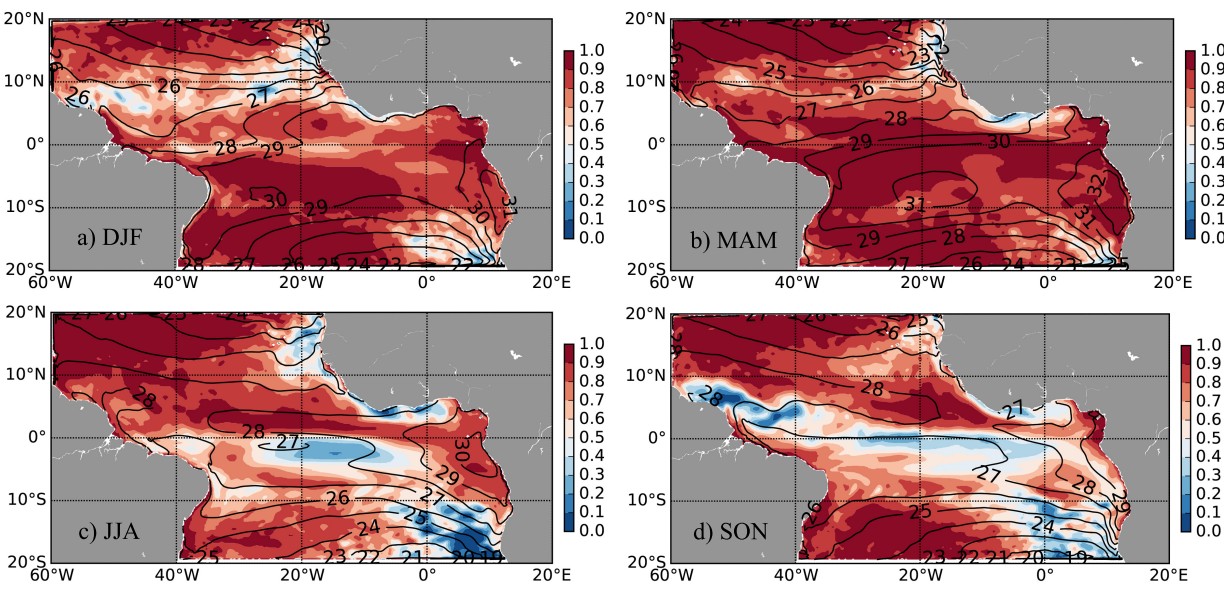

5 **Figure 6.** Coefficient of determination ($R^2$) between BIASED and BIASED-$\tau_{clim}$ seasonal SST time series, as in Figure 4.

