# Peer review of "Equatorial Atlantic interannual variability and its relation to dynamic and thermodynamic processes"

_Earth System Dynamics, 2017_

## Referee Comment (RC1) · Anonymous Referee #1 · 22 Jul 2017

Summary: This paper considers a set of 4 regional ocean-ABL simulations with climatological extratropical ocean conditions and a variety of free atmosphere conditions. Two of the simulations are "reference" REF simulations, one with observed winds (REF) and one with climatological winds (REF-CLIM). The second pair of simulations try to reproduce the mean biased state typical of CMIP-class models, and again use one interannually-variable wind set (BIASED) and one climatological wind set (BIASED-CLIM)(although both based on coupled model fields).

I will comment on my ratings above. I rate the interdisciplinarity as only good, since the model used is not a fully coupled system. I rate the science merits as only good

because the questions posed are very specific rather than generally applicable. I rate the technical quality as good, because I think the method can be improved upon (see below).

The primary claim of the paper is that the Tropical Atlantic variability in REF is absent in REF-clim, thus a wind-induced variability is responsible in contrast to a recent study by Nnamchi et al. [2015]. They go a bit further to insist that REF & REF-clim are more realistic than BIASED & BIASED-CLIM in terms of the mean state and basic variability. BIASED & BIASED-CLIM seem to be more similar to one another, which suggests that under these winds it is interannual thermodynamic forcing that might dominate the variability.

I find these claims and the experimental design moderately convincing that within the context of this model framework this analysis applies. However, a few important points should be clarified before publication.

1) I do not find these experiments convincing in the more general sense that coupled climate models necessarily behave like the BIASED runs and the real world behaves like the REF runs. This should be more clearly noted in the conclusion–i.e., a different evaluation framework is required in the coupled system is required (e.g., overwriting the coupled wind stress or coupled thermodynamic fluxes with a climatology), although the BIASED results here are suggestive. It certainly seems clear from this result that a biased mean state can affect ocean sensitivity & response.

2) Also, there is one flaw I see in the design of this experiment. The thermodynamic forcing–particularly sensible and latent heat fluxes–depend sensitively on the winds in the boundary layer. No attempt has been made to assess how much of the wind-stress change is actually realized as a thermodynamic forcing–not including the whole model, but just including the flux-windstress relationships–in the cases studied here. Within the mixed layer heat budget, I'd like to see what fraction of the For Nino-For Nina difference is explained this way. The hypothesis here is that a dynamical re-

sponse to winds is what changes, but maybe the thermodynamic response to winds is what actually changes. For an example of such a diagnosis, see Bates et al. (2012, http://dx.doi.org/10.1175/JCLI-D-11-00442.1). To do a much more careful analysis of this problem, a third member of each run set (REF-climthermo) could be run where the thermodynamic flux formulae only use the climatological winds but the stresses still rely on the interannual winds.

---

## Referee Comment (RC2) · Anonymous Referee #2 · 21 Aug 2017

The paper by Jouanno & colleagues investigates the role of thermodynamics and dynamics in the Equatorial Atlantic interannual variability, namely Atlantic Nino events. They use a suite of well-designed simplified ocean-ABL experiments together with a mixed layer budget to analyze the different contributions in the heat content budget.

The paper is well presented and relatively easy to follow. The design of the numerical simulations for the present study is appropriate, and the addition of the mixed layer budget is necessary to draw conclusions about the physics of the interannual variability. I recommend the paper for publication, however only after the authors revised several parts of the analysis and their interpretation.

[Figure]

1) The numerical simulations attempt to explore the role of dynamics by replacing the interannual wind variability by its climatology. I find myself wondering how much does the temperature/moisture adjust to this change? The conclusion that dynamics is the number one factor in setting the variability cannot be concluded this way. I would like to see the ML budget analyzed for the REF-tauclim, and most importantly how much is the thermodynamics forcing changed between the two simulation due to the changing winds.

2) The explanation of the ML budget is a bit confusing to me. I might have missed this somewhere but the contribution from the forcing to the ML tendency is positive, yet the authors keep saying that it acts as a damping (and vice versa for the mixing). Looking at the plots, I am not entirely convinced by the explanations for either set of runs (REFs or BIASEDs). It might be a wording issue, but this needs to be clarified.

3) The difference between Ninos and Ninas events are fairly small in all the plots (especially in the BIASED runs - about 1m change in isotherms!). We cannot clearly distinguish the changes between the events when the seasonal cycle is so dominant. The authors should concentrate on the variability, remove the seasonal cycle, this might give us confidence that the changes seen between the events are significant, and a better way to interpret the changes in the ML terms.

Some minor comments:

* introduction line 33: should be Bjerknes

* simulations page 2 line 33-35: which variables are forced? any issues due to the adjustment of the flow?

* simulations page 3: see comment 1) - T and q must adjust to the wind, so there is a thermodynamical adjustment due to the changes in dynamics which might mask the true response.

* page 4: Qns and Qs definitions are cryptic, please specify which fluxes they represent; also make sure you use subscripts where appropriate

* page 4, line 31: long living = long-lasting ?

* page 5, line 10: can you precise what is an acceptable range in term of variance to be more quantitative.

* pages 5-7: see major comments 1, 2 3 above. In addition, you might want to avoid the use of "most probably", "could also" when building your argument.

---

## Referee Comment (RC3) · Anonymous Referee #3 · 27 Aug 2017

Review for manuscript esd-2017-58 "Equatorial Atlantic interannual variability and its relation to dynamic and thermodynamic processes" by Jouanno et al.

The authors analyze different regional model runs to determine the relative contribution of the dynamic and thermodynamic forcing to interannual variability of the equatorial Atlantic sea surface temperature.

The manuscript is well written and addresses relevant scientific questions within the scope of Earth System Dynamics. The substantial and novel scientific results about the role of the thermodynamic forcing of interannual climate variability in the tropical Atlantic are timely and present a valuable contribution to the community interested in

tropical climate dynamics.

I think that the manuscript should be publishable after the authors have addressed my concerns mentioned below. My major remarks are (1) the results of a recently published study by Planton et al. 2017 (Main processes of the Atlantic Cold Tongue Interannual Variability, Climate Dynamics, published online.) are very similar to the results of the REF run discussed in section 3. Many aspects, such as a mixed-layer heat balance analysis for Atlantic Nino and Nina events, ocean-atmosphere fluxes during these events, are discussed in detailed in that paper but remain unreferenced here. (2) section 2.3 explaining in detail how the biased runs were constructed is hard to follow. I would appreciate a more detailed explanation of how the forcing for these runs was constructed. (3) I would appreciate if the authors could also include a discussion about predictability in their discussion and conclusion section. What do the new findings mean for the predictability of interannual climate variability in the tropical Atlantic by start of the art climate models?

I am offering some details to the remarks above and some minor remarks below.

Page 3, Line 13: Insert "out" after carried.

Page 3, sentence in lines 19-21 sounds strange: I would suggest changing it to: "This strategy allows to specifically remove the dynamical contribution of the interannual winds. However, thermodynamic contributions of wind variability (i.e. latent and sensible heat) are allowed to vary interanually."

Page 3, 3rd paragraph of section 2.3: I find it difficult to follow this paragraph and I would appreciate if additional information on how the forcing for BIASED and BIASED-tauclim runs were constructed. E.g., were the atmospheric variables taken directly from the coupled model (CNRM-CM5) and interpolated onto your grid? Was anything else done with the forcing? How does the CNRM-CM5 model compare to other CMIP 5 models, particularly in respect to the SST bias?

Page 3, Line 36, mixed layer balance equation: the meridional advection term (2nd term on the right hand side) is wrong. It should say: -âŇľv·∂_y TâŇł.

Page 5, line 10, "...but at levels that remain in an acceptable range": What is an acceptable range? I find this statement very subjective and suggest removing it.

Page 5, line 10, "s.t.d.": The abbreviation is not introduced. Also, I find the sentence in which the abbreviation is mentioned hard to understand. Please explain "interannual monthly s.t d."?

Page 5, whole sentence starting in line 24: The fact that net atmospheric heat flux anomalies are of elevated during cold ACT events and reduced during warm ACT events have been noted earlier. E.g. Planton et al, Main processes of the Atlantic Cold Tongue Interannual Variability, Climate Dynamics, published online, 2017, discuss this in some detail.

Page 5, Paragraph from line 27 to 38: The results presented here are convincing. However, they are very similar to the results presented by Planton et al., 2017, who also analyze the mixed layer heat balance during Atlantic Nino and Nina events. In addition, they discuss interannual wind anomalies (i.e. taux) in March to May being relevant for Atlantic Nino, Nina events that seem to agree with the results shown in your figure 3.

Page 6, line 18, "than in REF." sounds wrong. I think it should say "as in REF".

Page 7, line 8: add an "s" to suggest.

Page 7, line 9: no comma before "that"; "mode" must be capitalized.

Page 6-7, discussion and conclusion section: After reading the conclusions I was wondering about predictability. If in coupled climate models the dynamic response is too weak, then short term (1-3 month) predictability originating from equatorial wave propagation would be underestimated, right? Could the authors include some statements about predictability in their conclusions? After all, this is a major focus of the PREFACE

project.

---

## Author Comment (AC1) · 11 Oct 2017

Dear referee We acknowledge your careful reading and the comments you made on the manuscript "Equatorial Atlantic interannual variability and its relation to dynamic and thermodynamic processes" for publication in Earth System Dynamics. We have addressed all the comments and tried to incorporate your suggestions in the revised manuscript. You will find below the detailed reply to your reviews. Best regards Julien Jouanno on behalf of the authors.

Please also note the supplement to this comment:

[Figure]

https://www.earth-syst-dynam-discuss.net/esd-2017-58/esd-2017-58-AC1-supplement.pdf

[Figure]

**Supplement:**

Dear editor and referees

We acknowledge your careful reading and the comments you made on the manuscript "Equatorial Atlantic interannual variability and its relation to dynamic and thermodynamic processes" for publication in Earth System Dynamics. We have addressed all the comments and tried to incorporate your suggestions in the revised manuscript.

You will find below the detailed reply to the three reviews.

Moreover, note that B. Deremble has been removed from the co-authors list and passed in the acknowledgement. He considers his contribution was not enough to justify co-authorship.

Best regards
Julien Jouanno on behalf of the authors.

**Anonymous Referee #1**

Summary: This paper considers a set of 4 regional ocean-ABL simulations with climatological extratropical ocean conditions and a variety of free atmosphere conditions. Two of the simulations are "reference" REF simulations, one with observed winds (REF) and one with climatological winds (REF-CLIM). The second pair of simulations try to reproduce the mean biased state typical of CMIP-class models, and again use one interannually-variable wind set (BIASED) and one climatological wind set (BIASED-CLIM) (although both based on coupled model fields).

I will comment on my ratings above. I rate the interdisciplinarity as only good, since the model used is not a fully coupled system. I rate the science merits as only good because the questions posed are very specific rather than generally applicable. I rate the technical quality as good, because I think the method can be improved upon (see below).

The primary claim of the paper is that the Tropical Atlantic variability in REF is absent in REF-clim, thus a wind-induced variability is responsible in contrast to a recent study by Nnamchi et al. [2015]. They go a bit further to insist that REF & REF-clim are more realistic than BIASED & BIASED-CLIM in terms of the mean state and basic variability. BIASED & BIASED-CLIM seem to be more similar to one another, which suggests that under these winds it is interannual thermodynamic forcing that might dominate the variability.

I find these claims and the experimental design moderately convincing that within the context of this model framework this analysis applies. However, a few important points should be clarified before publication.

1) I do not find these experiments convincing in the more general sense that coupled climate models necessarily behave like the BIASED runs and the real world behaves like the REF runs. This should be more clearly noted in the conclusion–i.e., a different evaluation framework is required in the coupled system is required (e.g., overwriting the coupled wind stress or coupled thermodynamic fluxes with a climatology), although the BIASED results here are suggestive. It certainly seems clear from this result that a biased mean state can affect ocean sensitivity & response.

Our main point was to investigate whether a biased ocean mean state could affect the inter-annual variability of the sea surface temperatures in the Equatorial Atlantic. And we show that this is the case. Moreover, our results suggest that the wrong ocean mean state could be mainly induced by the wrong representation of the wind-stress forcing from the atmosphere. We would expect from a fully coupled set of simulations to reach the same conclusions. Indeed, Goubanova et al. (in preparation and personal com.) use a coupled-model framework in which they prescribe the observed wind-stress over the equatorial Atlantic. They show that the bias of the ocean mean state (ocean subsurface from 0 to 200m) is significantly reduced. The coupled model used in their study is very similar to CNRM-CM5, the one used in the present study. Other study, Sanchez-Gomez et al. (personal communication) used the anomaly coupling technique applied to the coupled model CNRM-CM5 to show that improving the wind-stress climatology over the Tropical Atlantic leads to a significant improvement of the equatorial variability (ATL3). Unfortunately there studies are not published. Below we show the figure by Sanchez-Gomez et al. which represents the mean state and the variability of the ATL3 SST in different coupled experiments: CTL (control run, free model), AC (anomaly coupling with prescribed wind-stress observed climatology), and ACI (iterative anomaly coupling). ORAS4 reanalysis is used here as reference. In the AC technique, only the climatology is corrected, letting free the anomalous component.

[Figure]

**Figure R0:** Seasonal cycle of SST mean and standard deviation averaged over the ATL3 region for the observations, CTL, AC and ACI experiments.

We rephrased the last paragraph of the conclusion as follows: "From a climate modelling perspective, and although a set of fully coupled simulations would be required to confirm our findings, our results are suggestive that a reduction of the mean and seasonal model biases in the Tropical Atlantic (in particular from the atmospheric component) would strongly benefit to the representation of the interannual variability."

2) Also, there is one flaw I see in the design of this experiment. The thermodynamic forcing–particularly sensible and latent heat fluxes–depend sensitively on the winds in the boundary layer. No attempt has been made to assess how much of the wind- stress change is actually realized as a thermodynamic forcing–not including the whole model, but just including the flux-windstress relationships–in the cases studied here. Within the mixed layer heat budget, I'd like to see what fraction of the For Nino-For Nina difference is explained this way. The hypothesis here is that a dynamical response to winds is what changes, but maybe the thermodynamic response to winds is what actually changes. For an example of such a diagnosis, see Bates et al. (2012, http://dx.doi.org/10.1175/JCLI-D-11-00442.1). To do a much more careful analysis of this problem, a third member of each run set (REF-climthermo) could be run where the thermodynamic flux formulae only use the climatological winds but the stresses still rely on the interannual winds.

As already mentioned in the manuscript (see Section 2.3), and as the reviewer propose, the heat and freshwater surface fluxes in simulations REF-$\tau_{clim}$ and BIASED-$\tau_{clim}$ remain forced by the interannual data (including inter-annual winds), so there is no thermodynamic response (i.e. changes of sensible and latent heat fluxes) directly due to the modified wind stress. By doing so, we are able to isolate the dynamical response to interannually varying winds. The third member you propose would allow to get more insight of the thermodynamic influence of the varying winds. Nevertheless, even if this question is very interesting, we think it is out of the scope of our study whose main objective was to specifically clarify/highlight the role of the ocean dynamics in controlling the interannual variability of the tropical surface temperatures.

**Anonymous Referee #2**

The paper by Jouanno & colleagues investigates the role of thermodynamics and dynamics in the Equatorial Atlantic interannual variability, namely Atlantic Nino events. They use a suite of well-designed simplified ocean-ABL experiments together with a mixed layer budget to analyze the different contributions in the heat content budget.

The paper is well presented and relatively easy to follow. The design of the numerical simulations for the present study is appropriate, and the addition of the mixed layer budget is necessary to draw conclusions about the physics of the interannual variability. I recommend the paper for publication, however only after the authors revised several parts of the analysis and their interpretation.

1) The numerical simulations attempt to explore the role of dynamics by replacing the interannual wind variability by its climatology. I find myself wondering how much does the temperature/moisture adjust to this change? The conclusion that dynamics is the number one factor in setting the variability cannot be concluded this way. I would like to see the ML budget

analyzed for the REF-tauclim, and most importantly how much is the thermodynamics forcing changed between the two simulation due to the changing winds.

It is not clear to us why the reviewer suggests that the predominance of the mechanical forcing cannot be concluded from our methodology. First of all, the heat and freshwater surface fluxes in simulation REF-$\tau_{clim}$ are forced by the interannual data (including inter-annual winds), so the sensible and latent heat fluxes are still function of interannual winds. This means that the direct influence of the wind on the thermodynamic forcing is kept interannual. We agree that the air-temperature, humidity (together with latent and sensible heat fluxes) will adjust but only as a consequence to the changes in mechanical input of momentum to the ocean surface (i.e. through modification of the SST in response to dynamical processes). We carefully read our result and discussion sections and we do not see any conclusion that would be invalidated by the fact that temperature/moisture adjust to the removal of the interannual variability of the mechanical input.

The ML budget analyzed for the REF-Tclim simulation is shown below as Figure R1. It does not bring much more additional information: the seasonal cycle and main balance between the terms are very close to REF and the differences between Nino and Nina years are much weaker than REF. This was expected since removing the interannual variability of the wind stress strongly weakens the interannual variability of the ACT temperatures weak [Fig 2a]. So we prefer to not include/discuss this budget in the manuscript.

2) The explanation of the ML budget is a bit confusing to me. I might have missed this somewhere but the contribution from the forcing to the ML tendency is positive, yet the authors keep saying that it acts as a damping (and vice versa for the mixing). Looking at the plots, I am not entirely convinced by the explanations for either set of runs (REFs or BIASEDs). It might be a wording issue, but this needs to be clarified.

Indeed, the contribution of the air-sea fluxes to the ML tendency remains positive all the year, which means that the atmospheric heat fluxes act to warm the ocean. However, we notice that during Niño years (i.e. warm anomalies), the warming by the air-sea fluxes is reduced compared to Niña years. This implies that air-sea fluxes are not the first order driver of the interannual anomalies in the cold tongue area (if it would be the case we would expect larger air-sea warming during Nino years).
We rephrase as follow to try to avoid confusion: "In contrast and as noted earlier in Planton et al. [2017], the warming by air-sea fluxes (FOR) and horizontal advection (HOR) is increased during cold events and reduced during warm events, so these processes act to reduce the temperature anomalies."

3) The difference between Ninos and Ninas events are fairly small in all the plots (especially in the BIASED runs - about 1m change in isotherms!). We cannot clearly distinguish the changes between the events when the seasonal cycle is so dominant. The authors should concentrate on the variability, remove the seasonal cycle, this might give us confidence that the changes seen between the events are significant, and a better way to interpret the changes in the ML terms.

As suggested, we did the exercise to remove the seasonal cycle. This is shown in Figure R2 in this reply. It provides additional information on the fact that Niño and Niña events are almost symmetrical in terms of amplitude, phase and processes (as already shown by Lubbecke and McPhaden 2017, GRL). But our feeling is that keeping the seasonal cycle helps to the interpretation. First, because it provides information on the sign of the different contributions to the ML budget. Second, because it allows to show that the interannual anomalies are weak compared to the seasonal cycle, a specificity of the Tropical Atlantic compared to the Tropical Pacific where Niño/Niña are much larger than the seasonal variability. The "1m change" in the thermocline depth between the warm and cold events in BIASED is informative by itself. It highlights that the surface warm/cold anomalies in BIASED are not due to changes in the thermocline depth.

Some minor comments:
* introduction line 33: should be Bjerknes
This has been corrected.

* simulations page 2 line 33-35: which variables are forced? any issues due to the adjustment of the flow?
The horizontal velocity, temperature, salinity and sea level are specified at the lateral boundaries. This is now mentioned. All the simulations are run from 1958 to 2015 and only data from 1979 to 2015 are analyzed. Owing to the small size of the domain (20°S-20°N), the 21 years spin-up is long enough for the upper ocean flow to adjust.

* simulations page 3: see comment 1) - T and q must adjust to the wind, so there is a thermodynamical adjustment due to the changes in dynamics which might mask the true response.
It is not clear to us what the referee call "true response". When removing the interannual variability of the dynamical forcing, most of the interannual variability of the SST in the cold tongue area is removed. We agree that there is an indirect modification of the air-sea fluxes in response to the changes in the dynamics (due to the adjustment of T and q and also SST), but since they

are the result of an adjustment we do not see how they could contribute to the removal of the interannual variability of the SST and avoid our conclusions.

* page 4: Qns and Qs definitions are cryptic, please specify which fluxes they represent; also make sure you use subscripts where appropriate * page 4, line 31: long living = long-lasting ?
We now precise the fluxes they represent : Qns (now renamed as Q*) represent the non penetrative part of the air-sea fluxes and Qs represents the penetrative part of the air-sea fluxes.
"Long living" has been replaced by "long lasting". Thanks.

* page 5, line 10: can you precise what is an acceptable range in term of variance to be more quantitative.
This has been removed.

* pages 5-7: see major comments 1, 2 3 above. In addition, you might want to avoid the use of "most probably", "could also" when building your argument.
We rephrased when required.

**Anonymous Referee #3**

Review for manuscript esd-2017-58 "Equatorial Atlantic interannual variability and its relation to dynamic and thermodynamic processes" by Jouanno et al.

The authors analyze different regional model runs to determine the relative contribution of the dynamic and thermodynamic forcing to interannual variability of the equatorial Atlantic sea surface temperature.

The manuscript is well written and addresses relevant scientific questions within the scope of Earth System Dynamics. The substantial and novel scientific results about the role of the thermodynamic forcing of interannual climate variability in the tropical Atlantic are timely and present a valuable contribution to the community interested in tropical climate dynamics.

I think that the manuscript should be publishable after the authors have addressed my concerns mentioned below. My major remarks are (1) the results of a recently published study by Planton et al. 2017 (Main processes of the Atlantic Cold Tongue Interannual Variability, Climate Dynamics, published online.) are very similar to the results of the REF run discussed in section 3. Many aspects, such as a mixed-layer heat balance analysis for Atlantic Nino and Nina events, ocean-atmosphere fluxes during these events, are discussed in detailed in that paper but remain unreferenced here. (2) section 2.3 explaining in detail how the biased runs were constructed is hard to follow. I would appreciate a more detailed explanation of how the forcing for these runs was constructed. (3) I would appreciate if the authors could also include a discussion about predictability in their discussion and conclusion section. What do the new findings mean for the predictability of interannual climate variability in the tropical Atlantic by stat of the art climate models?

Following your suggestions we complete the manuscript :
1) The paper by Planton et al. [2017] was not available when we submitted this manuscript, but we add reference to their study and discussion when required (in Introduction, Result, and Discussion-Conclusion Sections).
2) We provide more details on how the biased simulations were built.
3) We add a discussion on predictability (see details below)

I am offering some details to the remarks above and some minor remarks below.

Page 3, Line 13: Insert "out" after carried.
Inserted. Thanks.

Page 3, sentence in lines 19-21 sounds strange: I would suggest changing it to: "This strategy allows to specifically remove the dynamical contribution of the interannual winds. However, thermodynamic contributions of wind variability (i.e. latent and sensible heat) are allowed to vary interanually."
Indeed it sounds better. Thanks.

Page 3, 3rd paragraph of section 2.3: I find it difficult to follow this paragraph and I would appreciate if additional information on how the forcing for BIASED and BIASED-tauclim runs were constructed. E.g., were the atmospheric variables taken directly from the coupled model (CNRM-CM5) and interpolated onto your grid? Was anything else done with the forcing? How does the CNRM-CM5 model compare to other CMIP 5 models, particularly in respect to the SST bias ?.

We now provide additional information on the methodology used to build the BIASED forcing.
CNRM-CM5 model exhibit a marked equatorial Atlantic warm SST bias typical of the CMIP5 ensemble mean warm bias [Richter et al. 2008, Voldoire et al. 2014]. This was already mentioned.

Page 3, Line 36, mixed layer balance equation: the meridional advection term (2nd termontherighthandside) is wrong. Itshouldsay:-âNl'v·∂_yTâNł.
Thanks, this has been corrected.

Page 5, line 10, "...but at levels that remain in an acceptable range": What is an acceptable range? I find this statement very subjective and suggest removing it.
This has been removed.

Page 5, line 10, "s.t.d.": The abbreviation is not introduced. Also, I find the sentence in which the abbreviation is mentioned hard to understand. Please explain "interannual monthly s.t d."?
The sentence has been replaced by "interannual standard deviation"

Page 5, whole sentence starting in line 24: The fact that net atmospheric heat flux anomalies are of elevated during cold ACT events and reduced during warm ACT events have been noted earlier. E.g. Planton et al, Main processes of the Atlantic Cold Tongue Interannual Variability, Climate Dynamics, published online, 2017, discuss this in some detail.
We add reference to Planton et al. [2017] when discussing this point.

Page 5, Paragraph from line 27 to 38: The results presented here are convincing. However, they are very similar to the results presented by Planton et al., 2017, who also analyze the mixed layer heat balance during Atlantic Nino and Nina events. In addition, they discuss interannual wind anomalies (i.e. taux) in March to May being relevant for Atlantic Nino, Nina events that seem to agree with the results shown in your figure 3.
Indeed, they are similar and we add reference to Planton et al. [2017]

Page 6, line 18, "than in REF." sounds wrong. I think it should say "as in REF". Page 7, line 8: add an "s" to suggest.
Corrected.

Page 7, line 9: no comma before "that"; "mode" must be capitalized.
Corrected.

Page 6-7, discussion and conclusion section: After reading the conclusions I was wondering about predictability. If in coupled climate models the dynamic response is too weak, then short term (1-3 month) predictability originating from equatorial wave propagation would be underestimated, right? Could the authors include some statements about predictability in their conclusions? After all, this is a major focus of the PREFACE project.

We agree and complete the discussion-conclusion section as follows :
  "From a climate modelling perspective, and although a set of fully coupled simulations would be required to confirm our findings, our results are suggestive that a reduction of the mean and seasonal model biases in the Tropical Atlantic (in particular from the atmospheric component) would strongly benefit to the representation of the interannual variability. The variability of the Atlantic Cold Tongue exerts a significant influence on the climate of the surrounding regions and more specifically on the West African monsoon [Okumura and Xie 2004, Caniaux et al. 2011] or on rainfall variability in the northeast of Brazil [Kushnir et al., 2006]. In terms of predictability at seasonal time scale of these phenomena, our results suggest that the ability of the climate models to maintain a realistic stratification and east-west tilt of the thermocline is key in correctly representing the response of the summer coupled system to spring wind anomalies."

[Figure]

**Figure R1:** As Figure 3 of the manuscript but using REFtauclim simulation. Here, Niño and Niña years were also selected using observed Atl3 SSTs from TropFlux.

[Figure]

**Figure R2.** Same as Figure 3 of the paper but shown as anomalies compared to the seasonal cycle.